# RLEF: Grounding Code LLMs in Execution Feedback with Reinforcement Learning

Jonas Gehring [1]  Kunhao Zheng [1]  Jade Copet [1]  Vegard Mella [1]  Taco Cohen [1]  Gabriel Synnaeve [1]

## Abstract

Large language models (LLMs) deployed as agents solve user-specified tasks over multiple steps while keeping the required manual engagement to a minimum. Crucially, such LLMs need to ground their generations in any feedback obtained to reliably achieve the desired outcomes. We propose an end-to-end reinforcement learning method for teaching models to leverage execution feedback in the realm of code synthesis, where state-of-the-art LLMs struggle to improve code iteratively compared to independent sampling. We benchmark on competitive programming tasks and achieve large performance gains with both small (8B parameters) and large (70B) models, outperforming previous work while reducing the number of samples required by an order of magnitude. Our analysis of inference-time behavior demonstrates that our method produces LLMs that effectively leverage automatic feedback over multiple steps.

## 1. Introduction

The consistent increase in capabilities of Large Language Models (LLMs) has prompted researchers and developers to benchmark and deploy them in increasingly complex environments (Brown et al., 2020; OpenAI, 2023; AI @ Meta, 2024). An emerging research direction is to employ LLMs as agents to solve tasks in multiple steps with little to no human oversight, querying external computation or data sources when needed or as dictated by manual scaffolding (Schick et al., 2023; Kapoor et al., 2024). Such autonomous use of LLMs is of interest for ensuring accurate answers to user queries with up-to-date information (Mialon et al., 2024), interaction with websites (Yao et al., 2022), or generating code to implement software fea-

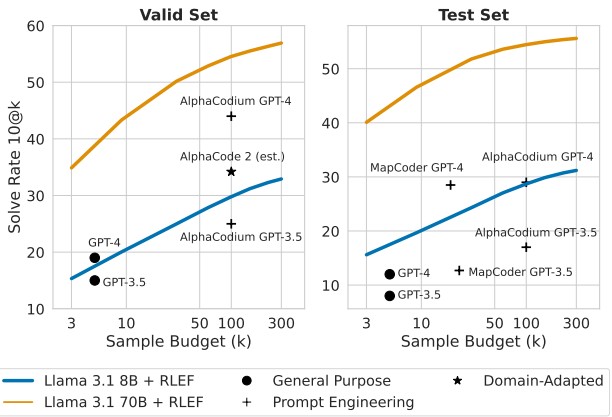

Figure 1: Solve rates of Llama 3.1 Models after RLEF training on CodeContests, compared to previously reported results across sampling budgets (log scale).

tures from high-level descriptions (Yang et al., 2024).

We posit that any decision-making agent offering a natural language interface has to possess two skills: First, the ability to accurately deduce a user's intent when prompted; for LLMs, this is typically achieved by fine-tuning to follow instructions according to user preferences (Ouyang et al., 2022; Rafailov et al., 2023). Second, feedback on intermediate results of the agent's actions has to be taken into account to arrive at the desired outcome. For example, a web page that contains a necessary bit of information might have gone offline, requiring another search engine query. In the context of code generation, feedback can provide information about implementation bugs as well as constraints that are inefficient or cumbersome to specify in full detail, e.g., software and hardware platform details or library dependencies. Intermediate feedback is therefore crucial to ground LLM generations in the concrete situations encountered at inference time.

In this work, we aim to endow pre-trained LLMs with the aforementioned skills – task alignment and grounding in inference-time feedback – in the domain of code synthesis from natural language descriptions (Chen et al., 2021; Rozière et al., 2023). Here, feedback is naturally provided as the result of the execution of generated code in the form of error messages and unit test results. However, to date,

[1]Meta FAIR. Correspondence to: Jonas Gehring <jgehring@meta.com>, Gabriel Synnaeve <gab@meta.com>.

*Proceedings of the 42nd International Conference on Machine Learning*, Vancouver, Canada. PMLR 267, 2025. Copyright 2025 by the author(s).

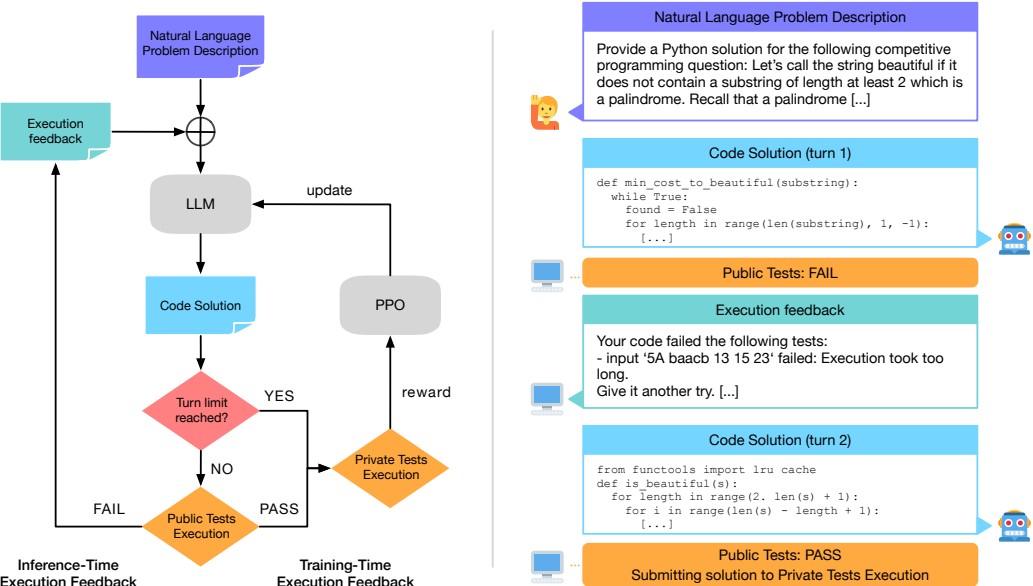

Figure 2: **Left:** Overview of reinforcement learning with execution feedback (RLEF). The LLM is repeatedly prompted to implement code according to a problem description. Each attempt is evaluated on a public test set; upon failure, feedback is inserted into the conversation. If public tests are passing, or a specified turn limit is reached, execution on additional, private tests determines the reward signal. The model is then updated to optimize the reward with PPO. **Right:** Example dialogue with two model responses. Execution feedback hints at an inefficient first solution, to which the model responds to utilizing a cache. The code passing the public test sets will be evaluated on the full test set.

utilizing such feedback for code generation with LLMs has failed to yield substantial improvements when taking computational demands into account; indeed, obtaining samples independently often results in higher accuracy for a fixed inference budget (Kapoor et al., 2024; Xia et al., 2024). As a test bed to investigate and improve grounding in execution feedback, we frame code generation as an iterative task, repeatedly asking an LLM to produce code according to a provided natural language description (Figure 2). After each generation, code is evaluated on example test cases, and the resulting feedback is provided as additional context for subsequent attempts. We thus obtain an interactive environment where actions correspond to code and observations correspond to execution feedback. Importantly, such a framing permits end-to-end optimization with reinforcement learning (RL) algorithms to maximize a reward signal – here, a binary reward based on whether the final code solution passes a set of held-out test cases.

We benchmark our training method incorporating repeated code actions and execution feedback in a reinforcement learning context (RLEF) on CodeContests (Li et al., 2022), a challenging competitive programming benchmark. Starting from Llama 3.1 models (AI @ Meta, 2024), we achieve substantial performance improvements, surpassing previous state-of-the-art results while reducing the number of generations required by an order of magnitude (Figure 1). Our analysis shows that RLEF training unlocks the capability to leverage inference-time machine feedback, rendering

LLMs effective in iterative, multi-turn scenarios. Our improvements from RLEF on CodeContests further generalize to HumanEval+ and MBPP+, two popular benchmarks for code synthesis, and to increased sample budgets compared to training time.

## 2. Method

### 2.1. Iterative Code Synthesis

We structure the code synthesis task as a multi-turn conversation in which an LLM is repeatedly prompted to generate a code solution to a natural language problem description. After each solution, we provide an automatically generated response with results obtained by executing the solution's code against test cases. This setup is applicable to language models tuned for the use case of interacting with users in a chat format and follows previous work on self-repair for code generation (Shinn et al., 2023; Olausson et al., 2024).

Crucially, we utilize two different sets of test cases: a *public* test yields execution feedback that can be accessed during repeated attempts and forms the basis of selecting a final solution, whereas a *private* test set ultimately determines the correctness of the final solution. Separate test sets provide two main benefits. First, if test inputs and outputs are fixed, held-out tests guard against shortcuts during the optimization procedure in which an LLM can copy expected test outputs in subsequent answers, based on ex-

ecution feedback. Second, running a full test suite may be computationally demanding, and a limited set of public tests can accelerate the iterative code generation procedure. It may, however, be desirable to maximize test coverage for execution feedback at inference time, and we verify that this can indeed improve performance (Appendix B.3).

Our conversation flow for code generation is depicted in Figure 2. Concretely, we start the dialogue with the problem description and query the LLM for an initial solution. The solution is verified against the public test set, which yields results in the form of passed and failed test cases, as well as potential syntax or runtime errors. If any public test fails, this execution feedback is formatted and appended to the dialogue. The LLM is then queried for an updated code solution, with the original problem text, previous solutions, and their respective feedback provided in the prompt. If the solution passes all public tests, or a specified turn limit is reached, it is considered final and will be submitted for evaluation on the private test set. We refer to Appendix C for prompt and execution feedback templates.

## 2.2. Reinforcement Learning with Execution Feedback

The iterative code synthesis described in the previous section can be seen as a Markov Decision Process (MDP), and the language model as a policy (Sutton & Barto, 2018). For generality, we assume a partially observable MDP, as our reward function utilizes a held-out, private test set that is not accessible to the policy (unless an exact textual representation of the desired program behavior is provided in the problem description). Observations and actions are provided as tokenized text sequences. Concretely, the initial observation $o_0$ is the problem description, and actions $a_t$ at each step $t$ are textual responses. Successive observations $o_t$ consist of past observations and actions, including execution feedback obtained by evaluating the previous action $a_{t-1}$ on public tests. Episodes end when public test evaluation succeeds or a specified step limit is reached. At the end of an episode, a scalar reward is provided corresponding to whether all public and private tests are passing. We do not use reward discounting (i.e., $\gamma = 1$).

For optimizing a policy in the above environment, we employ Proximal Policy Optimization (PPO), a common choice for fine-tuning large language models (Schulman et al., 2017; Ziegler et al., 2020; Ouyang et al., 2022). Following previous work, we include a KL penalty in our reward signal, acting both as an entropy bonus and as regularization towards the distribution of the LLMs we start from. In initial experiments, we found that a possible failure mode concerns the generation of invalid code in non-final responses, which we address by providing a small penalty for invalid responses. Denoting the policy to be optimized with $\pi$ and the initial policy with $\rho$,

and abbreviating previous observations and actions with $c_t = o_0, a_0, o_1, a_1, \ldots, o_t$, our rule-based reward function at step $t$ is:

$$R(s_t, a_t) = r(s_t, a_t) - \beta \log \frac{\pi(a_t|c_t)}{\rho(a_t|c_t)}, \text{ where}$$

$$r(s_t, a_t) = \begin{cases} 1, & \text{if end of episode and all tests pass} \\ -1, & \text{if end of episode and any test fails} \\ -0.2, & \text{if } a_t \text{ does not contain valid code} \end{cases}$$

with a constant $\beta$ trading off between task reward and KL maximization. For PPO, we compute policy gradients by incorporating a concurrently learned value function as a baseline, i.e., we train the policy to maximize the advantage $A_t = -V(c_t) + \sum_{i=t}^{T} R(s_i, a_i)$; see Appendix A.1.

We note that while the above MDP considers full responses as actions, the underlying policy and value functions are implemented as language models outputting single tokens. Selecting a suitable action space for optimization hence requires consideration in our setup, and a suitable choice may depend on the concrete task at hand. We propose to model the policy at the token level while learning a value function for whole turns; compared to optimizing both models at either the turn or token level, this hybrid approach worked best in our early experiments. Hence, we predict the value of a response $a_t$ from the last token of its respective prompt, and we use a single advantage value for each token action within a response. Our response-based value estimation is closely related to Zhou et al. (2024); however, we do not train an additional Q-function. For the KL penalty, we found it beneficial to compute the probabilities of responses $\pi(a_t|c_t)$ as the geometric mean rather than the product of token probabilities. This counteracts a possibly detrimental bias towards shorter generations, in particular for non-final responses.

## 3. Experimental Results

### 3.1. Setup

We perform experiments on the CodeContests benchmark (Li et al., 2022), which requires generating a code solution to a problem specified in natural language along with a textual description of public test cases. Problems are of high difficulty and used in human competitive programming with a focus on algorithms, data structures, and runtime efficiency. The correctness of solutions is evaluated with private tests that are hidden from contestants; hence, we present feedback from public tests only. CodeContests consists of a training set and two evaluation sets, "valid" and "test", with 117 and 165 problems, respectively; we use the former for model and hyperparameter selection. We optimize our models on the training set, from which we discard 669 of the 13,328 problems due to missing

Table 1: Results on CodeContests of our initial and RLEF-trained models compared to prior work. The sample budget $k$ in $n@k$ refers to the number of LLM responses, e.g., 1@3 for our results corresponds to a single rollout with up to three model responses. The 70B model obtains state-of-the-art results after RLEF, and significantly outperforms AlphaCodium and MapCoder generally, and on the test set with a fraction of the samples. The RLEF-trained 8B model outperforms AlphaCodium with 100 samples and MapCoder (gpt-3.5-turbo) with 3 samples.

| Model | Source | n@k | Valid Set | Test Set |
|---|---|---|---|---|
| AlphaCode 9B | Li et al. (2022) | 10@1000 | 16.9 | 13.3 |
| AlphaCode 41B + clustering | Li et al. (2022) | 10@1000 | 21.0 | 16.4 |
| Code Llama 34B + PPO | Xu et al. (2024) | 10@1000 | 19.7 | 22.4 |
| AlphaCodium gpt-3.5-turbo-16k | Ridnik et al. (2024) | 5@100 | 25 | 17 |
| AlphaCodium gpt-4-0613 | Ridnik et al. (2024) | 5@100 | 44 | 29 |
| AlphaCodium Llama 3.1 70B Instruct | Ours | 5@100 | 34.2 | 27.8 |
| MapCoder gpt-3.5-turbo-1106 | Islam et al. (2024) | 1@23 | - | 12.7 |
| MapCoder gpt-4-1106-preview | Islam et al. (2024) | 1@19 | - | 28.5 |
| Llama 3.0 8B Instruct | Ours | 1@3 | 4.1 | 3.2 |
| + RLEF | Ours | 1@3 | 12.5 | 12.1 |
| Llama 3.1 8B Instruct | Ours | 1@3 | 8.9 | 10.5 |
| + RLEF | Ours | 1@3 | 17.2 | 16.1 |
| Llama 3.1 70B Instruct | Ours | 1@3 | 25.9 | 27.5 |
| + RLEF | Ours | 1@3 | **37.5** | **40.1** |
| Llama 3.1 8B Instruct | Ours | 10@100 | 21.7 | 24.8 |
| + RLEF | Ours | 10@100 | 29.8 | 28.7 |
| Llama 3.1 70B Instruct | Ours | 10@100 | 50.2 | 50.3 |
| + RLEF | Ours | 10@100 | **54.5** | **54.5** |

public or private test cases. We train all models to output Python code solutions.

The Llama 3 family of models (AI @ Meta, 2024) comprises our initial policies, specifically the Instruct 8B and 70B parameter models of the 3.0 and 3.1 release. These models exhibit strong code generation performance out of the box and are able to follow instructions in the prompt, alleviating the need for an initial fine-tuning stage prior to RL training. During training and for evaluations, unless noted, we set the turn limit to allow for 3 LLM attempts at solving each problem. We perform 12,000 and 8,000 updates to the 8B and 70B models, respectively, and select checkpoints based on valid set performance. Hyperparameters and further experimental details are provided in Appendix A.

We follow Li et al. (2022) in reporting results as $n@k$ average solve rates. The $n@k$ metric represents the expectation that any of $n$ solutions, selected from $k$ samples in total, is correct, i.e., passes all tests. In our multi-turn setup, each turn counts as a sample. This allows for fair comparisons with respect to sample budgets, which is particularly relevant when employing large LLMs with high inference cost in agentic scaffoldings (Kapoor et al., 2024)[1].

[1]For simplicity, we consider a full LLM response as a single sample in our evaluations. Note that for iterative code generation, the allocated sample budget may not be fully utilized as a success-

### 3.2. Main Results

In Table 1, we list solve rates on the CodeContests valid and test sets for iterative code generation with up to three turns, along with previously reported results. When sampling from our models, we use temperatures 0.2 for 1@3 and 1.0 for 10@100, and nucleus sampling with top-p 0.95 in all cases (Holtzman et al., 2020). Each solve rate is estimated on 200 rollouts, using the estimator described by Li et al. (2022). We compare against AlphaCode (Li et al., 2022) and PPO with rewards from test execution on the Code Llama 34B model from Xu et al. (2024), both of which report results with a large number of samples. AlphaCodium (Ridnik et al., 2024) and MapCoder (Islam et al., 2024) are high-performing agentic frameworks built on top of the proprietary GPT models and combine chain-of-thought prompting, code execution, program repair, and, in the case of AlphaCodium, automatic test generation.

With RLEF training, we improve markedly on the original Llama 3.1 models. Notably, on the test set, the 70B model beats AlphaCodium with GPT-4 with a single rollout compared to 5 solutions from 100 samples (38.0 and 29). Likewise, the 8B model after RLEF is slightly ahead compared to the similar-sized AlphaCode 9B model (16.1 and 13.3), but with a sample budget of 3 in our case and

ful public test run will result in early termination of a dialogue.

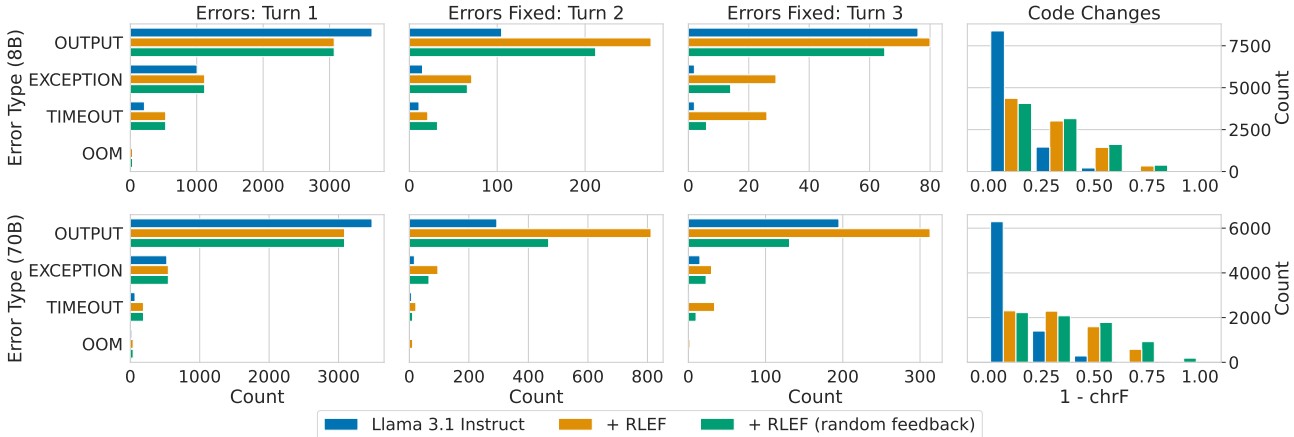

Figure 3: Behavior analysis of initial and RLEF-trained models with respect to public test results for 8B (top) and 70B (bottom) models. Within 20 rollouts per problem (5640 in total), we count errors in the initial solution (turn 1); errors turned into correct code in turns 2 and 3; and code changes across successive solutions according to the chrF metric. RLEF-trained models make fewer errors initially, can fix errors more reliably, and perform larger code edits; initial models frequently repeat previous solutions. With random execution feedback, error recovery is severely impaired.

1,000 for AlphaCode. While we cannot compare directly to the more recent AlphaCode 2, a performance estimate of 34.2 on the valid set for 10@100 puts our 70B model ahead (37.5) with just 3 samples[2]. When considering a larger budget of 100 samples, corresponding to 33 rollouts, the 70B Instruct model beats previously reported results, including AlphaCodium on the valid set. Evaluating AlphaCodium with the 70B model results in lower scores under a similar budget. With RLEF, we obtain further improvements to 54.5 on the valid and test set. The relative gains over the initial models, while still significant, are reduced in the 10@100 setting as compared to the 1@3 setting. Kirk et al. (2024) observe that RL training of LLMs can reduce the diversity of outputs, and we interpret our results as further evidence of their hypothesis.

Table 1 also highlights that the released Llama 3.1 models offer competitive performance on CodeContests from the start, which we attribute to a focus on coding capabilities during instruction tuning (AI @ Meta, 2024). However, our method is also highly effective on the previously released 3.0 8B model, improving 1@3 solve rates on both the valid (4.1 → 12.5) and test (3.2 → 12.1) set. Thus, RLEF may be useful as a partial substitute for instruction tuning for tasks where ground-truth rewards from automatic evaluation are available.

[2]AlphaCode Team (2023) train and evaluate on non-disclosed competition problems but report a sample efficiency increase of 10,000x over AlphaCode, which achieves a 10@1M solve rate of 34.2 on the valid set.

### 3.3. Inference-time Behavior

In Table 2, we first take a closer look at single- and multi-turn performance with a fixed budget of 3 LLM generations (1@3). This corresponds to our iterative setup with up to three model responses, or three independent responses for single-turn results. We further consider generalization to two popular code generation benchmarks, HumanEval+ and MBPP+ (Liu et al., 2023a), which we modify to match our iterative code generation setup with "base" tests for inference-time execution feedback and "plus" tests for solve rate estimation (see Appendix C.4 for details). Our results demonstrate that, when considering a fixed sample budget, base models rarely benefit from access to faulty solutions and execution feedback in the multi-turn code generation setup. This also applies to gpt-4o-2024-05-13, which shows stronger performance when sampling solutions independently on CodeContests and HumanEval+. After RLEF training, the 8B and 70B Llama 3.1 models both benefit from execution feedback and can therefore achieve larger gains on top of improved single-turn scores, with the exception of the 8B model on CodeContests and MBPP+ where single-turn performance drops. While multi-turn gains from RLEF are most pronounced on CodeContests, the training domain of our models, we also observe notable improvements on HumanEval+ and MBPP+. For confidence bounds that confirm the robustness of our results, as well as evaluation on LiveCodeBench, see Appendix B.1.

Next, we seek to determine where the gains of RLEF training stem from. Based on the improved single-turn results in Table 2, we hypothesize that, for the 70B model, these are partly due to training on the specific domain of competitive programming questions. More importantly, higher

Table 2: 1@3 solve rates in single-turn (ST) and multi-turn (MT) setups for base and RLEF models from the Llama 3.1 (L3.1) series. On CodeContests, iterative code generation yields modest gains at best and drops in performance at worst, unless RLEF training is employed. Improvements from RLEF on CodeContests in the multi-turn setting carry over to HumanEval+ and MBPP+, which require slightly different execution feedback formatting.

| Model | CC. Test | | HE+ | | MBPP+ | |
|---|---|---|---|---|---|---|
| | ST | MT | ST | MT | ST | MT |
| GPT-4o | 25.3 | 24.3 | 82.8 | 80.7 | 68.8 | 71.7 |
| L3.1 8B Inst. | 11.8 | 10.5 | 65.3 | 63.9 | 58.3 | 60.5 |
| + RLEF | 9.7 | 16.1 | 67.5 | 69.5 | 57.0 | 63.1 |
| L3.1 70B Inst. | 26.2 | 27.4 | 73.2 | 75.0 | 66.9 | 70.2 |
| + RLEF | 30.1 | 40.1 | 78.6 | 80.4 | 67.6 | 72.2 |

scores in the iterative setting for both the 8B and 70B model could be attributed to either an increased capability of sampling diverse solutions within a rollout or more targeted self-repair based on execution feedback. For probing the sensitivity of our models to the observed feedback, we perform inference-time ablations with *random* execution feedback by executing a faulty solution to an unrelated problem, but still end the dialogue if the current solution passes public tests (details in Appendix C.2).

In Figure 3, we consider errors on public tests (to which the execution feedback relates) over 20 rollouts on the valid and test set combined. After RLEF, both the 8B (top row) and 70B (bottom row) models produce fewer wrong outputs in their initial response but are more prone to exceeding the allocated time limit. In subsequent responses, recovery from all error categories is significantly improved. With random feedback, however, we see a clear impairment of self-repairs, demonstrating that RLEF allows LLMs to effectively leverage the provided feedback. We further gauge changes from one response to the next by computing the character n-gram F-Score (Popović, 2015, chrF) among successive codes (Figure 3, right). This underscores a shortcoming of the Instruct models without RLEF in that they perform only minimal code edits; indeed, we observe that they frequently output the same code solution despite inline feedback pointing out errors.

The analysis above in Figure 3 reveals two key findings about RLEF. 1. Samples within a rollout are of higher diversity (less similar codes). 2. Edits are also targeted, resulting in fewer successful repairs with random execution feedback. This finding is echoed in Figure 4a, in which we compare models with true and *random* feedback across different turn limits. Here, we compare pass@1 and pass@10 metrics, irrespective of different sample budgets due to varying turn limits (Chen et al., 2021). While pass@1 captures the precision with which we arrive at a

Table 3: 1@3 solve rates starting from Llama 3.1 models. Comparison of different methods for acquiring iterative code synthesis capabilities. RLEF is the most effective training method, followed by supervised fine-tuning (SFT). We find few-shot prompting detrimental to Instruct models.

| Method | 8B Instruct | | 70B Instruct | |
|---|---|---|---|---|
| | Valid | Test | Valid | Test |
| – | 8.9 | 10.5 | 25.9 | 27.5 |
| Few-Shot Prompting | 8.5 | 8.5 | 22.5 | 20.3 |
| SFT | 10.3 | 10.0 | 27.7 | 27.2 |
| RLEF | **17.2** | **16.1** | **37.5** | **40.1** |

correct final solution, pass@10 reflects the ability to recall a correct solution (i.e., whether any of 10 solutions passes the private tests). On both valid and test sets, random feedback results in a drop in pass@1, which is further amplified as the turn limit increases. This provides further evidence for less targeted repair capabilities with random feedback, as programs can be repaired less reliably. Notably, with ground truth feedback, the pass rate keeps increasing with higher turn limits. For pass@10, the difference between true and random execution feedback is less pronounced. As pass@10 can be optimized by sampling many diverse candidate solutions within a dialogue, these results indicate that with random feedback, our models resort to sampling a succession of diverse, potentially correct solutions.

Finally, we evaluate the generalization across turn limits with respect to a given sample budget. In Figure 4b, we perform rollouts with temperature 1.0 to emphasize performance at higher sample budgets by increasing the diversity of generations. We compute $10@k$ solve rates by distributing $k$ samples equally across rollouts with different turn limits. For the 8B model (top row), prior to RLEF training, best performance can be obtained with independent samples (1 turn), with the exception of the test set above 30 samples. The initial 70B model performs better with 3 or 5 turns; although, for small budgets, single-turn performance is competitive. After RLEF, we observe that 3, 5, and 10 turns yield a consistent improvement over independent sampling, with best performance obtained with 5 turns. In all cases, increasing the turn limit to 10 provides no benefits under a fixed sample budget.

### 3.4. Ablation Studies

3.4.1. LEARNING ITERATIVE CODE SYNTHESIS

We investigate whether LLMs can, apart from our RL training, be effective in multi-turn code generation using few-shot prompting (Brown et al., 2020) and supervised fine-tuning (SFT). Lacking suitable ground truth training examples for SFT, we mine rollouts on the CodeContests training set with Llama 3.1 70B Instruct and filter them based on

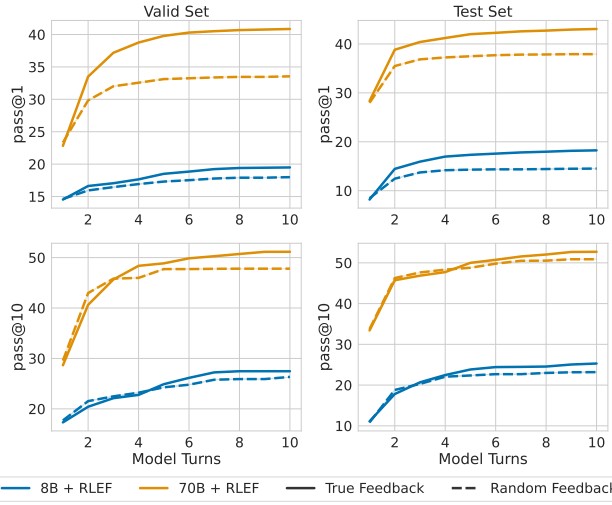

(a)

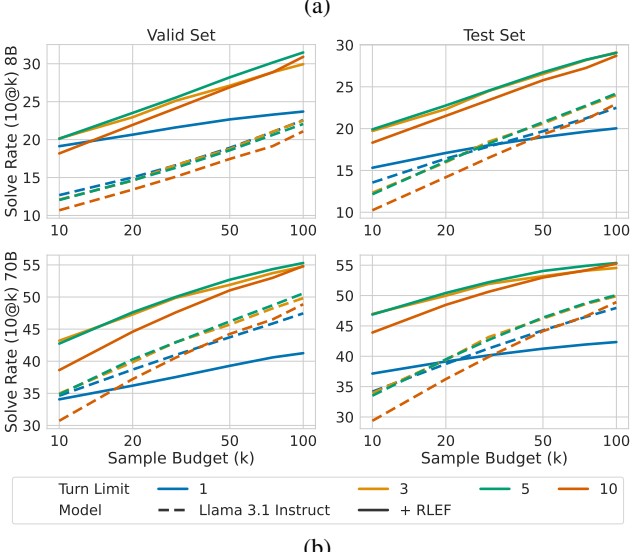

(b)

Figure 4: **(a)** Pass@1 and pass@10 across turn limits with RLEF-trained models, providing either true or random execution feedback (temperature 0.2). With random feedback pass@1 is reduced while pass@10 suffers only slightly, indicating that programs can be repaired less consistently. **(b)** Impact of turn limits on 10@k solve rates per sample budget (top: 8B model, bottom: 70B model) with temperature 1.0. With RLEF, iterative code generation can leverage up to 5 turns to achieve compute-optimal performance.

the correctness of final solutions. We then fine-tune Base and Instruct versions of the Llama 3.1 8B and 70B parameter models on the mined corpus and also source it for few-shot examples (Appendix A.3). The results in Table 3 show that few-shot prompting is detrimental to the instruction-tuned models. In Appendix B.2, we report few-shot 1@3 solve rates for pre-trained models and find that they achieve lower performance compared to zero-shot prompting for instruction models (1.2 and 1.8 for 8B, 4.6 and 5.8 for 70B on valid and test set, respectively). SFT improves Instruct

Table 4: 1@3 solve rates starting from Llama 3.1 models, temperature 0.2. Conventional single-turn (ST) compared to our multi-turn (MT) training method. MT training yields larger improvements compared to ST.

| Model | RLEF Training | Valid | | Test | |
|---|---|---|---|---|---|
| | | ST | MT | ST | MT |
| 8B Instruct | – | 9.4 | 8.9 | **11.6** | 10.5 |
| | ST | 10.3 | 10.2 | 9.9 | 10.9 |
| | MT | **16.2** | **17.2** | 9.5 | **16.1** |
| 70B Instruct | – | 25.6 | 25.9 | 25.9 | 27.5 |
| | ST | **28.3** | 31.1 | 27.3 | 32.9 |
| | MT | 25.8 | **37.5** | **30.1** | **40.1** |

model performance on the validation set only; we do not see improvements on the test set. For pre-trained models, we see improvements from SFT but lower scores compared to instruction-tuned models (Appendix B.2). With RLEF, we obtain significantly higher solve rates compared to SFT models, underscoring the efficacy of our RL training loop.

### 3.4.2. SINGLE-TURN TRAINING

In Table 4, we compare our iterative code generation setup to traditional, single-turn generation where the model is not presented with inference-time feedback. We use the same training loop for single generations, albeit without the penalty for invalid code (Section 2.2), as this is subsumed by the reward signal for incorrect solutions. For Llama 3.1 8B Instruct, single-turn training (ST) hurts performance on the test set. The 70B model benefits from single-turn training and improves over multi-turn SFT results in Table 3. Moreover, we observe transfer in that applying the single-turn model in a multi-turn setting improves 1@3 solve rates. We attribute this to the existent but comparably weak multi-turn capabilities of the vanilla 70B Instruct model. Overall, we see strongest performance with the RLEF method employing multiple turns at training and inference time.

Further ablations are in the appendix. In Appendix B.4 we evaluate the effect of training a dedicated repair model on outputs of the single-turn 8B training run in Table 4, similar to Le et al. (2022). Together, the single-turn and repair model obtain 1@3 solve rates of 14.8 on the valid set and 12.6 on the test set; an improvement over the single-turn model alone (10.2 and 10.9) but significantly below the corresponding multi-turn model (17.2 and 16.1). In Appendix B.5, we show that withholding public test execution feedback during training results in significantly worse performance. Finally, in Appendix B.6, we experimentally validate the design choice of a turn-level value function (Section 2.2).

# 4. Related Work

**Code Generation from Natural Language** A large body of work explores code synthesis from natural language descriptions, often leveraging large quantities of source code for pre-training and instruction fine-tuning (Clement et al., 2020; Chen et al., 2021; Austin et al., 2021; Li et al., 2023; Gunasekar et al., 2023; Rozière et al., 2023; AI @ Meta, 2024). This approach has led to notable performance gains on popular benchmarks.

**Inference-Time Repair and Scaffolding** Recent research shows that techniques such as prompt scaffolding and flow engineering (Ridnik et al., 2024; Islam et al., 2024; Zhang et al., 2024) can improve performance on challenging benchmarks but incur high inference costs by chaining multiple LLM calls. Other work focuses on re-prompting and verifying code through execution (Shinn et al., 2023; Chen et al., 2024b; Zhong et al., 2024). However, recent studies suggest that independent sampling of solutions is competitive and more efficient (Olausson et al., 2024; Kapoor et al., 2024) when considering equal sampling budgets. With our method, the self-repair capabilities of LLMs can be dramatically enhanced, resulting in superior performance of iterative code generation for both small and large sample budgets. At the same time, we propose to trade complex, domain-specific prompt engineering and scaffolding for domain-specific reinforcement fine-tuning.

**Fine-Tuning/RL with Execution Feedback** Fine-tuning LLMs with reinforcement learning has become a widely adopted strategy for aligning outputs to desired targets, relying on specialized reward models or automatic signals (Ziegler et al., 2020; Touvron et al., 2023; OpenAI, 2023; DeepSeek-AI et al., 2024; AI @ Meta, 2024). For code synthesis, this reward signal can be derived from executing generated solutions against test cases (Shojaee et al., 2023; Dou et al., 2024; Yu et al., 2024). Specifically, Le et al. (2022) fine-tune an LLM with policy gradients and next-token loss on execution rewards. To improve performance at test time, they train dedicated models for outcome prediction and program repair ("critic sampling"), albeit without incorporating execution outputs in the prompt. Liu et al. (2023b) extend this work with a fine-grained reward function. Xu et al. (2024) fine-tune a stronger, code-specific LLM in a simpler setup with a binary reward from unit tests and observe substantial improvements from RL on CodeContests. In contrast, we expand the natural-language-to-code setting to an iterative environment where execution feedback is not only provided as a scalar reward but also in textual form. This allows a single model to learn both code synthesis and code repair capabilities without relying on ground-truth solutions or extra inference scaffolding.

**Concurrent Approaches and Longer-Horizon RL** Con-

currently, Kumar et al. (2024) propose a two-stage RL method (SCoRe) to improve self-correction by generating two successive solutions. In contrast to our method, SCoRe does not leverage execution feedback at inference time and instead asks the model to reconsider its initial solution. While this approach suits domains lacking automatic feedback, it cannot benefit from grounding via execution information. Furthermore, inference-time feedback can help the model generalize to new environments after training. Chen et al. (2024a) address code generation with human feedback and develop an appropriate SFT strategy by training a separate code repair model. In our work, we effectively leverage automatically generated feedback, formatted in natural language, with a single model only. More recently, DeepSeek-AI et al. (2025) observe emerging reasoning capabilities with a large-scale application of GRPO (Shao et al., 2024) to math and code problems and achieve high performance on competitive programming tasks. We thus consider the training of reasoning models with program execution feedback and, likewise, the introduction of execution feedback to math domains, a promising avenue for future research.

Past work on applying reinforcement learning to LLMs on longer-horizon decision-making tasks placed an emphasis on the necessary grounding in the environment, such as text-based navigation games (Carta et al., 2023), text games (versus an oracle LLM) and web-shopping (Zhou et al., 2024), and visual observations (Zhai et al., 2024). While our work follows similar motivations, we address a fundamentally different domain, code synthesis, which features a significantly larger action space compared to previous work, i.e., the space of valid Python programs.

# 5. Conclusion

**Limitations.** While our results demonstrate effective usage of inference-time feedback, the code synthesis task we consider is limited to improving a single solution to a given problem. Generalizing our method to environments with larger tasks that require decomposition, either via manual scaffolding or, eventually, in a self-directed manner, remains the subject of further research. Iterating on the execution results of unit tests naturally requires test cases, which may not be readily available. We regard a potential combination with automatic unit test generation (Watson et al., 2020; Jain et al., 2024a) as an interesting avenue for further experiments.

In this work, we proposed reinforcement learning from execution feedback (RLEF), a fine-tuning method for LLMs that endows them with a crucial capability for autonomous operation: grounding future generations in environment feedback. We applied RLEF to iterative code synthesis and obtained substantial improvements in solve rates on the

CodeContests competitive programming benchmark while reducing the required sample budget for inference. The RLEF-trained models further generalize to increased turn limits and to HumanEval+ and MBPP+, two popular code generation benchmarks that exhibit simpler programming questions and different execution feedback formatting. Our in-depth analysis revealed that, while an increase in correct first-turn generations and in the diversity of successive generations offers a major contribution to performance, our models also meaningfully take execution feedback into account and resolve errors over multiple turns.

**Acknowledgements.** We thank Quentin Carbonneaux for significant contributions to this work, as well as Chris Cummins, Olivier Duchenne, Fabian Gloeckle, Baptiste Roziere, Sten Sootla, Nicolas Usunier, and Sida Wang for helpful technical contributions, suggestions, and insightful discussions.

## Impact Statement

Successful grounding of LLMs for code generation execution feedback will amplify their utility when applied to impactful tasks such as assisting software development and performing quality control. In general, however, increasing the capabilities of LLMs, now widely deployed in a range of applications, requires quality control and guard-railing to promote safety and minimize potentially harmful output. We limit our study to the generation of source code, where we confine the execution of model-generated output to local sandboxes. We believe the framework of Shavit et al. (2023) regarding the governance of AI agents to be a useful resource for practitioners.

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

# A. Experimental Details

## A.1. RLEF

We initialize separate policy and value function networks from pre-trained and instruction-tuned LLMs as indicated in the respective experiments; for the value function, we replace the output layer with a randomly initialized linear projection. For PPO, we use AdamW (Loshchilov & Hutter, 2019) with a learning rate of $2e^{-7}$, weight decay of 0.1, and a linear warm-up over 50 steps. We set the KL regularization factor $\beta$ of the reward term to 0.05 (Section 2.2). All models are trained with an online, asynchronous training infrastructure that decouples inference and optimization. We incorporate importance sampling in PPO's clipped surrogate objective (Schulman et al., 2017, Eq.7):

$$r_t(\theta) = \frac{\pi_\theta(a_t|c_t)}{\pi_{\theta_{\mathrm{old}}}(a_t|c_t)} \operatorname{stop\_grad}\left(\min\left(\frac{\pi_\theta(a_t|c_t)}{\pi_{\mathrm{b}}(a_t|c_t)}, 1\right)\right),$$
$$L^\pi(\theta) = \hat{\mathbb{E}}_t\left[\min\left(r_t(\theta)\hat{A}_t, \operatorname{clip}\left(r_t(\theta), 1-\epsilon, 1+\epsilon\right)\hat{A}_t\right)\right]$$

for model parameters $\theta$, normalized advantage $\hat{A}_t$, and the behavior policy $\pi_{\mathrm{b}}$. We set $\epsilon = 0.2$.

For optimizing the value function, we use a clipped value loss. With value model parameters $\psi$ and reward function $R(s_t, a_t)$ (see Section 2.2) we have

$$R_t = \sum_{i=t}^{T} \gamma^{i-t} R(s_i, a_i),$$
$$L^V(\psi) = \hat{\mathbb{E}}_t\left[\frac{1}{2}\max\left((V_\psi(c_t) - R_t)^2, (\operatorname{clip}(V_\psi(c_t), V_{\psi_{\mathrm{old}}}(c_t) - \alpha, V_{\psi_{\mathrm{old}}}(c_t) + \alpha) - R_t)^2\right)\right]$$

where we set the discount factor $\gamma$ to 1 and the value clipping threshold $\alpha$ to 0.2.

During training, we perform inference with a temperature of 1.0; we use neither nucleus (top-p) nor top-k sampling. We collect 1024 rollouts and perform 4 updates on 256 sequences each. Models are evaluated every 800 updates, and we select the final model based on validation set performance. We train our models on NVidia H100 GPUs; a training run takes approx. 20 wall time hours. With the above parameters we use 288 (128 for training, 160 for inference) and 2304 (1024 for training, 1280 for inference) GPUs for 8B and 70B models, respectively.

## A.2. Code Execution

We evaluate candidate solutions with the accompanying code-base of Li et al. (2022)[3] using Python 3.10. All problems in the validation and test set specify a memory limit, and only a few problems define a time limit. If specified, we apply these limits for RLEF training and evaluations; otherwise, we use a 1GB memory limit and maximum wall clock time of 10 seconds per test case.

## A.3. Supervised Fine-Tuning

We perform supervised fine-tuning (SFT) for the ablations in Section 3.4.1. In order to assemble a training dataset, we perform iterative code generation with our proposed setup on the CodeContests training set with the Llama 3.1 70B Instruct model. We set top-p to 0.95 and sample a temperature for each response in $U(0.1, 1.0)$. For each problem in the training set we collect 100 multi-turn rollouts and obtain 313,639 successful trajectories.

We fine-tune models for next-token prediction, computing losses on the last response only (i.e., on responses passing both public and private tests); this produced slightly better models compared to training on all responses. We sweep over learning rates $5e^{-6}$ and $2e^{-6}$, and 2 and 3 epochs with a batch size of 64 and sequence length 8192. A linear warmup is performed over 10 steps, and learning rates are annealed according to a cosine schedule. Weight decay is set to 0.1. Models are evaluated after 200 optimizer steps with AdamW and we select final parameters based on validation set performance.

---

[3]`https://github.com/google-deepmind/code_contests`

Table 5: 1@3 solve rates with 95% confidence intervals on tasks from Table 2 as well as LiveCodeBench up to 10/2024.

| Model | Setting | CC. Test | HumanEval+ | MBPP+ | LiveCodeBench |
|---|---|---|---|---|---|
| GPT-4o | single-turn | 25.3 (23.2-26.3) | 82.8 (81.8-83.2) | 68.8 (68.1-69.2) | 49.9 (48.7-50.0) |
|  | multi-turn | 24.3 (23.2-25.4) | 80.7 (79.8-81.7) | 71.7 (71.4-72.2) | 48.4 (48.0-49.0) |
| Llama 3.1 8B Instruct | single-turn | 11.6 (11.3-12.0) | 65.3 (63.8-66.2) | 58.3 (57.5-58.7) | 22.5 (21.6-22.7) |
|  | multi-turn | 10.5 (10.1-10.8) | 63.9 (62.8-65.1) | 60.4 (59.7-60.9) | 23.1 (22.6-23.5) |
| + RLEF | single-turn | 9.7 (9.5-10.0) | 67.5 (66.7-68.0) | 57.0 (56.4-57.2) | 22.7 (22.4-22.9) |
|  | multi-turn | 16.1 (15.8-16.4) | 69.5 (68.6-70.5) | 63.1 (62.7-63.5) | 27.8 (27.4-28.1) |
| Llama 3.1 70B Instruct | single-turn | 26.2 (25.6-26.6) | 73.2 (72.5-73-9) | 66.9 (66.1-67.4) | 38.1 (37.3-38.4) |
|  | multi-turn | 27.4 (27.0-27.8) | 75.0 (74.3-76.0) | 70.2 (69.7-70.8) | 40.9 (40.4-41.3) |
| + RLEF | single-turn | 30.1 (29.7-30.4) | 78.7 (78.1-79.0) | 67.6 (67.2-67.8) | 36.9 (36.5-37.0) |
|  | multi-turn | 40.1 (39.7-40.4) | 80.4 (79.7-81.0) | 72.2 (71.9-72.5) | 42.4 (42.2-42.7) |

Table 6: **(a)** 1@3 solve rates for few-shot prompting and supervised fine-tuning (SFT) with Llama 3.1 Base models on CodeContests. **(b)** Further results from Llama 3.1 Instruct 8B (1@3): withholding execution feedback from public training during RL; learning a value function on the token level; training a dedicated code repair model and applying it to outputs of the single-turn RL model.

| Model | Method | Valid | Test |
|---|---|---|---|
| 8B Base | Few-Shot | 1.2 | 1.8 |
|  | SFT | 6.9 | 3.5 |
| 70B Base | Few-Shot | 4.6 | 5.8 |
|  | SFT | 11.1 | 10.9 |

(a)

| Method (8B) | Valid | Test |
|---|---|---|
| RLEF | 17.2 | 16.1 |
| No Execution Feedback | 12.2 | 10.9 |
| Token-level Value Function | 13.1 | 13.7 |
| Single-turn RL | 10.2 | 10.9 |
| Single-turn w/ Repair | 14.8 | 12.6 |

(b)

# B. Additional Experimental Results

## B.1. Extended Results on Generalization

Table 2 lists performance of RLEF-trained models and GPT-4o on benchmarks that are not strictly within the competitive programming domain. We report confidence intervals for all results, along with results on LiveCodeBench (Jain et al., 2024b), in Table 5. We select LiveCodeBench questions up to October 2024.

We follow Li et al. (2022) in computing 95% confidence intervals for n@k results: given an evaluation with N samples, we draw N results with replacement and estimate 1@3 solve rates, repeat this 200 times, and select the 2.5 and 97.5 percentiles as lower and upper bounds of the confidence interval.

Table 5 shows that gains from RLEF-training on CodeContests on other tasks are statistically significant in our evaluation setting, where we draw 20 multi-turn samples per model and task combination. For single-turn results, we consider the first generation of each sample only. For consistency, we use 200 samples on CodeContests (except for GPT-4o) as in Table 1. Confidence intervals for GPT-4o on CodeContests are therefore wider.

## B.2. Pre-trained Models

In Table 6a we list solve rates for few-shot prompting and supervised fine-tuning from pre-trained Llama 3.1 models. We observe significantly lower performance compared to the Instruct models in all cases (Table 3).

### B.3. Feedback from Private Tests

Our main evaluations on CodeContests match our training setting, i.e., we provide inference-time feedback on public test cases and estimate solve rates on private (and the dataset's generated) tests. The number of public test cases in the CodeContests validation and test sets vary between 1-7, with a median of 1; typically, a higher number of private tests and a large number of generated tests are available per problem.

We verify whether our RLEF-trained models can benefit from larger test sets during inference by including feedback from private and generated tests. Specifically, we test each model response against 20 available test cases, including private tests, and provide execution feedback for up to 8 failed test cases. Comparing 1@3 solve rates (temperature 0.2) with a turn limit of 3, the 8B RLEF model can improve from 17.2 to 18.1 on the valid set, whereas on the test set we see a drop from 16.1 to 14.4. For the 70B RLEF model, validation set performance improves from 37.5 to 40.4, and on the test set we obtain 41.2 compared to 40.1 with feedback limited to public tests.

### B.4. Extra Repair Model

Le et al. (2022) implement program repair on top of an RL-trained LLM with two extra models: a "critic" predicts the joint outcome of all unit tests (e.g., success, failure, runtime error) and can be used for ranking and determining promising prefixes, and a "repair" model maps wrong solutions to ground truth solutions. In this spirit, we evaluate the effect of a dedicated repair model to improve the single-turn 8B model from Section 3.4.2 as follows.

During the RL training procedure, we collect all generations that do not pass the public unit tests. For the training duration of 12,000 gradient steps, this amounts to 1.48M samples. Next we construct training dialogues with the original prompt (as described in Appendix C.1), the wrong generation, and a random correct generation for the respective problem from the CodeContests training set. We apply additional processing to the CodeContests solutions by making sure they do indeed pass the provided unit tests and unifying their indentation. We then train repair models via supervised fine-tuning of Llama 3.1 8B Instruct, sweeping over learning rates $5e^{-6}$, $2e^{-6}$, and $1e^{-6}$, and 1 or 2 epochs with a batch size of 64 and a sequence length of 8192.

For evaluations, we estimate 1@3 solve rates by generating an initial program with the RL-trained model followed by up to two independent samples from the repair model. Similar to our main RLEF setting, we refrain from (further) repair if the latest solution passes the public tests. We evaluate all models from the sweep in intervals of 400 gradient steps and select the best checkpoint based on validation set performance. This checkpoint achieves, in combination with the single-turn RL model, a 1@3 solve rate of 14.8 on the validation set and 12.6 on the test set, which is a significant increase over the single-turn RL model alone (10.2 and 10.9, respectively; Table 4) but falls short of the corresponding RLEF-trained model which combines code synthesis and code repair (17.2 and 16.1, respectively; Table 1)

### B.5. RL Training Without Public Test Execution Feedback

We validate our setup consisting of inline execution feedback and early stopping based on public tests (Section 2.1) with an ablation where we withhold information from public tests. Concretely, we remove execution feedback from the prompt for subsequent solutions (Appendix C.1), starting directly with "Give it another try". We always ask the model for two follow-up solutions this way (i.e., for a total of three solutions). We do keep our reward definition from Section 2.2 but do not end episodes when public tests are passing.

The resulting model, starting from Llama 3.1 8B Instruct, obtains a 1@3 solve rate of 12.2 on the validation and 10.9 on the test set (Table 6b). This is better than the initial instruct model (8.9 and 10.2, respectively) but significantly below the corresponding RLEF-trained model (17.2 and 16.1, respectively).

### B.6. Token-Level Value Function

Here we do not train a value function the level of responses (Section 2.2, Appendix A.1) but rather predict a value for each token of a response. Our reward formulation remains unchanged; consequently, due to the discount factor being set to 1, the value function target (reward-to-go) for each token of a response is the same. However, we now compute separate per-token advantages.

With this approach and otherwise identical settings, we achieve a 1@3 solve rate of 13.1 on the validation and 13.7 on the test set, starting from Llama 3.1 8B Instruct (Table 6b). This is below the 17.2 and 16.1 results with the turn-level value

function (Table 1).

## C. Prompts

### C.1. CodeContests

In the initial prompt, we substitute `${problem}` by the original problem description as-is.

---
**Initial Prompt**

```
Provide a Python solution for the following competitive programming question: \${
problem}.

Your code should be enclosed in triple backticks like so: '''python YOUR CODE HERE
'''. Use the backticks for your code only.
```
---

In the execution feedback prompt below, we show templates for the four different error types we consider: wrong answer, exception, timeout, and out of memory. We then show the respective feedback for each failing test.

---
**Execution Feedback**

```
Your code failed the following tests:

- input '${input}' failed:
Expected output '${expected_output}' but got '${observed_output}'
- input '${input}' failed:
${stacktrace}
- input '${input}' failed: Execution took too long.
- input '${input}' failed: Out of memory.

Give it another try.
Your code should be enclosed in triple backticks like so: '''python YOUR CODE HERE
'''. Use the backticks for your code only.
```
---

### C.2. Random Feedback Ablation

In Section 3.3 we test RLEF-trained models with random execution feedback. For each problem, we sample a different problem from the respective test set that contains incorrect solutions. We obtain unrelated feedback by evaluating one of these incorrect solutions, chosen at random, against the corresponding public tests and present the resulting feedback to the model. If none of the incorrect solutions fail the public tests, we evaluate `raise NotImplementedError()`. In this case, the feedback will contain backtraces pointing to this error. Otherwise our dialog proceeds as usual, i.e., if the code solution produced by the LLM passes the true public tests of the problem in questions we stop and evaluate the solution on all test cases.

### C.3. Few-Shot Prompting

For the few-shot ablations in Section 3.4, we select successful trajectories from the Llama 3.1 70B Instruct model on problems from the CodeContests training set. We select trajectories with both 2 and 3 successful attempts to as demonstrations for successful multi-turn code generation. For instruction models, we initialize the dialog with the few-shot examples, separating them with an empty assistant message. For few-shot experiments with pre-trained models (Appendix B.2), we use a dialog format in which each message is either prefixed by `[USER]` or `[ASSISTANT]`. The token for `||`, an invalid symbol in Python, is used as a message delimiter.

### C.4. HumanEval+

HumanEval problem prompts consist of starter code, with a docstring and example tests following the function declaration.

---

**Initial Prompt**

```
Write a solution to the following problem and make sure that it passes the tests:
${problem}
```

---

We then provide the problem prompt again at the start of each model response for completion.

The tests in HumanEval+ consist of a single function with several `assert` statements. In order to obtain execution feedback for individual tests, we extract them from original test function (for computing pass rates, we use the original test code). We further transform `assert` statements into matching function calls of Python's built-in `unittest.TestCase` class. This way, test failures will result in more informative `AssertionError` exceptions with run-time values; these are provided as `assertion_error` to the template. We also show successful test cases.

---

**Execution Feedback**

```
Your code failed some test cases:

- Failure: `${test}`:
`${assertion_error}`
- Failure: `${test}`:
${stacktrace}
- Failure: `${test}`:
Execution took too long.
- Success: `${test}`

Give it another try.
```

---

## C.5. MBPP+

Each MBPP prompt consists of a problem description and a single example test.

---

**Initial Prompt**

```
Provide a Python solution for the following problem: ${problem}
Your code should pass these tests:

${test}

Your code should be enclosed in triple backticks like so: ```python YOUR CODE HERE
```. Use the backticks for your code only.
```

---

Execution feedback follows the HumanEval+ format from Appendix C.4 with additional formatting guidelines.

Execution Feedback

```
Your code failed some test cases:

- Failure: `${test}`:
`${error}`
- Failure: `${test}`:
${stacktrace}
- Failure: `${test}`:
Execution took too long.
- Success: `${test}`

Give it another try.
Your code should be enclosed in triple backticks like so: ```python YOUR CODE HERE
```. Use the backticks for your code only.
```

## D. Examples

The following examples are selected from the valid set of CodeContests with the RLEF-trained 70B model, using temperature 0.2 and top-p 0.95. We apply some light re-formatting of the initial prompts for better readability.

In the first example, the first model response is on the right track, but the first `print()` statement outputs a wrong value. The second response implements the requested algorithm naively, but the code is inefficient due to the loop from $l$ to $r$; it would take about two minutes for the last line in the public test. Finally, the third response contains code that is both correct and fast.

Initial Prompt

```
Provide a Python solution for the following competitive programming question: You are given two integers l and r, l ≤ r. Find
the largest possible value of a mod b over all pairs (a, b) of integers for which r ≥ a ≥ b ≥ 1.
As a reminder, a mod b is a remainder we get when dividing a by b. For example, 26 mod 8 = 2.

Input

Each test contains multiple test cases.
The first line contains one positive integer t (1 ≤ t ≤ 10^4), denoting the number of test cases. Description of the test
cases follows.
The only line of each test case contains two integers l, r (1 ≤ l ≤ r ≤ 10^9).

Output

For every test case, output the largest possible value of a mod b over all pairs (a, b) of integers for which r ≥ a ≥ b ≥ 1.

Example

Input

4
1 1
999999999 1000000000
8 26
1 999999999

Output

0
1
12
499999999

Note

In the first test case, the only allowed pair is (a, b) = (1, 1), for which a mod b = 1 mod 1 = 0.
In the second test case, the optimal choice is pair (a, b) = (1000000000, 999999999), for which a mod b = 1..
Your code should be enclosed in triple backticks like so: ```python YOUR CODE HERE ```. Use the backticks for your code only.
```

## Model Response

```python
t = int(input())

for _ in range(t):
    l, r = map(int, input().split())

    if r // 2 + 1 >= l:
        print((r // 2))
    else:
        print(r % l)
```

## Execution Feedback

```
Your code failed the following tests:

- input `4
1 1
999999999 1000000000
8 26
1 999999999
` failed:
Expected output `0
1
12
499999999
` but got `0
1
13
499999999
`

Give it another try.
Your code should be enclosed in triple backticks like so: ```python YOUR CODE HERE ```. Use the backticks for your code only.
```

## Model Response

```python
t = int(input())

for _ in range(t):
    l, r = map(int, input().split())

    max_mod = 0
    for b in range(l, r + 1):
        a = r
        mod = a % b
        max_mod = max(max_mod, mod)

    print(max_mod)
```

## Execution Feedback

```
Your code failed the following tests:

- input `4
1 1
999999999 1000000000
8 26
1 999999999
` failed:
Execution took too long

Give it another try.
Your code should be enclosed in triple backticks like so: ```python YOUR CODE HERE ```. Use the backticks for your code only.
```

## Model Response

```python
t = int(input())

for _ in range(t):
    l, r = map(int, input().split())

    if r == l:
        print(0)
    else:
        b = (r // 2) + 1
        if b >= l:
            print(r % (b))
        else:
            print(r % l)
```

## Result

Passes public and private tests.

In the second example, the first code solution does not reset the value for $p_k$. The second and correct attempt fixes this. Note that both solutions contain an unused variable, `curr` in the first and `curr_a` in the second response. For this question, an approach with dynamic programming that starts initializes $p_k$ from $p_{k-1}$ would be preferred; however, the second solution passes all test cases, too.

---

**Initial Prompt**

```
Provide a Python solution for the following competitive programming question: You have an array a consisting of n distinct
positive integers, numbered from 1 to n. Define p_k as $$$p_k = \sum_{1 \le i, j \le k} a_i mod a_j, where x \bmod y denotes the
remainder when x is divided by y. You have to find and print p_1, p_2, \ldots, p_n$$$.

Input

The first line contains n - the length of the array (2 ≤ n ≤ 2 · 10^5).
The second line contains n space-separated distinct integers a_1, ..., a_n (1 ≤ a_i ≤ 3 · 10^5, a_i ≠ a_j if i ≠ j).

Output

Print n integers p_1, p_2, ..., p_n.

Examples

Input

4
6 2 7 3

Output

0 2 12 22

Input

3
3 2 1

Output

0 3 5.
Your code should be enclosed in triple backticks like so: ```python YOUR CODE HERE ```. Use the backticks for your code only.
```

---

**Model Response**

```python
n = int(input())
a = list(map(int, input().split()))

p = 0
result = []
for k in range(n):
    curr = a[k]
    for i in range(k + 1):
        for j in range(k + 1):
            p += a[i] % a[j]
    result.append(p)

for num in result:
    print(num, end=" ")
```

---

**Execution Feedback**

```
Your code failed the following tests:

- input `4
6 2 7 3
` failed:
Expected output `0 2 12 22
` but got `0 2 14 36 `
- input `3
3 2 1
` failed:
Expected output `0 3 5
` but got `0 3 8 `

Give it another try.
Your code should be enclosed in triple backticks like so: ```python YOUR CODE HERE ```. Use the backticks for your code only.
```

## Model Response

```python
n = int(input())
a = list(map(int, input().split()))

result = []
total = 0
for k in range(n):
    curr_total = 0
    curr_a = a[:k+1]
    for i in range(k+1):
        for j in range(k+1):
            if i != j:
                curr_total += a[i] % a[j]
    result.append(curr_total)

for num in result:
    print(num, end=" ")
```

## Result

Passes public and private tests.

