# OpenReview forum: "RLEF: Grounding Code LLMs in Execution Feedback with Reinforcement Learning"
_ICML.cc/2025/Conference — ICML 2025 spotlightposter_

### Official Review · Reviewer_3JDi · 2025-03-10

**Overall Recommendation:** 3

**Summary:**

The paper proposes an end-to-end multi-turn RL framework to teach LLM self-repair/refine based on execution feedback, particularly focused on the code generation domain, where the unit tests and execution feedback is easy to obtain. The main algorithm is PPO with turn-level value function to calculate the advantage, along with a common KL penalty. Empirically, the method shows strong improvement on the CodeContests benchmark using Llama 3.1 models, and sees generalization performance across HumanEval and MBPP. Some interesting ablation studies are also conducted, such as few-shot prompting, SFT on success attempts, single-turn RLEF, etc.

**Claims And Evidence:**

Most of the claims the paper made are from the experiment results (section 3.2), centered around the effectiveness of the multi-turn rl based on execution feedback:
- the strong performance of RLEF compared with prior work, which supported by comparing with a wide range of baselines such as alphacode, code llama and alphacodium, etc.
- the generalization performance to HumanEval and MBPP, can we report CI for the numbers in Table 2 as well? The generalization performance seems worse compared to CC. Test, is the improvement statistically significant?
- RLEF tends to increase the errors fixed in the followup turns, and give much more code changes. This is examined in both the llama 7b and 80b models. Some questions here: (1). any hypothesis on why there is more timeout in later turns for the model being RLEFed. (2). for the eval of code changes (1-chrF), could we have a more refined analysis on model is simply re-attempting? or is doing real fixes?
- There are some claims about the diversity within a roll-out, L312-L316. I am a little bit confused here, i think diversity only values when (1). we want to do independent sampling; so we get a higher pass@k as k increases or (2). the diversity at the end of the roll-out, this is the response we will eventually used? In view of the diversity within the rollout, why this is a desirable criteria, i feel more evidence or justification is needed here.
- In ablation studies, the study of baselines (i.e., few-shot prompting and SFT) is interesting, any idea on why few-shot shows worse performance? is it a more general statement?
- Another claim regarding with the pre-trained models seem to benefit more from IF, than code-specific SFT for code generation performance, it would be give more hypothesis or justification here.

**Essential References Not Discussed:**

N/A

**Experimental Designs Or Analyses:**

The benchmark, baseline and evaluation metrics in general makes sense, some minor issues i saw:
- I do like the idea of testing how much the model learns from the execution feedback, but random feedback seems a too weak baseline, it would be ideal to add other more reasonable baselines, such as no execution feedback, give model binary / numerical reward only, or using a subset of public tests? An even more realistic setup in to let LLM self-generate some unit tests, and use it in the multi-turn training process, this might be out of the scope of the current paper, but might be interesting to see.

**Methods And Evaluation Criteria:**

The method of using end-to-end multi-turn RL to solve the code repair problem is pretty reasonable and solid. Code generation naturally has the execution feedback to be used as intermediate feedback to teach self-repair.

The evaluation is conducted on common code generation benchmarks, CodeContests, MBPP and HumanEval.

**Other Comments Or Suggestions:**

See discussions in previous sections.

**Other Strengths And Weaknesses:**

Strength:
- The paper studies an interesting problem of code-generation using multi-turn RL.
- The extensive set of ablation studies is interesting in this paper, such as the comparison with few-shot, SFT, single-turn RL.

Weakness:
- One major weakness I saw from the paper is that the method section is too dense, and it would be great if it could be written into more details, such as the turn-level value function, the task-specific implementations, etc.
- As mentioned before, it would be interesting to include more realistic case studies regarding to the quality / availability in the unit tests to be used in RLEF.

**Questions For Authors:**

See discussions in previous sections.

**Relation To Broader Scientific Literature:**

This paper is relevant to a broader community such as LLM, multi-turn RL, code generation.

**Theoretical Claims:**

There is no theoretical claims in the paper, N/A.

---

> ### Author Rebuttal · Authors · 2025-04-01
>
> We deeply appreciate the reviewer's comments and feedback on our manuscript.
>
> > the generalization performance to HumanEval and MBPP, can we report CI for the numbers in Table 2 as well? The generalization performance seems worse compared to CC. Test, is the improvement statistically significant?
>
> We will extend Table 2 with results on LiveCodeBench and confidence intervals for the Appendix: https://imgur.com/a/CpsZMJB
>
> CIs are estimated following AlphaCode, i.e., we repeatedly (200x) sample a subset of the solutions with replacement and estimate 1@3 solve rates. We then take the 2.5% and 97.5% percentiles.
>
> > any hypothesis on why there is more timeout in later turns for the model being RLEFed
>
> Our hypothesis is that this is an effect of solutions passing the public tests but failing under increased problem sizes that may be part of the private test sets.
>
> > could we have a more refined analysis on model is simply re-attempting? or is doing real fixes?
>
> > i think diversity only values when (1). we want to do independent sampling; so we get a higher pass@k as k increases or (2). the diversity at the end of the roll-out, this is the response we will eventually used? In view of the diversity within the rollout, why this is a desirable criteria, i feel more evidence or justification is needed here.
>
> Diversity of solutions within a single rollout can be important. When a proposed solution fails, it may be advantageous to start with a fresh/different approach rather than to address fixes, e.g., if the wrong algorithm was selected, fixing the runtime error would not lead it to be the correct one. Notably, our analysis shows that, without specific training, available LLMs are often not good at refining or changing their solutions.
>
> > In ablation studies, the study of baselines (i.e., few-shot prompting and SFT) is interesting, any idea on why few-shot shows worse performance? is it a more general statement?
>
> We don't claim that our conclusions regarding the negative effect of few-shot prompting can be readily generalized to other domains. However, we think the general ranking of learning methods as few-shot < SFT < RL is accurate.
>
> > I do like the idea of testing how much the model learns from the execution feedback, but random feedback seems a too weak baseline, it would be ideal to add other more reasonable baselines, such as no execution feedback, give model binary / numerical reward only, or using a subset of public tests?
>
> The choice of random execution feedback to test the sensitivity is motivated as follows: with our training regime, a trained model can infer that a solution is wrong solely based on the fact that it is prompted for another solution, and the public tests are already included in the initial prompt. Hence, a model insensitive to textual feedback could simply ignore it and propose another solution.
>
> It is thus hard to arrive at a pure "no execution feedback" setting for this experiment, since any re-prompting signals to the model that the previous solution was wrong. Nevertheless, we ran a small study where we replace the textual execution feedback with the string "Consider if the previous solution is correct and provide a new one if it is not." We see small drops in performance for both 8B (17.1 -> 16.8 on valid, 16.0 -> 14.5 on test) and 70B (37.1 -> 34.7, 40.6 -> 40.1) RLEF-trained models for 1@3 solve rates. We get a more pronounced performance reduction from random execution feedback (8B: 17.1 -> 16.0, 16.0 -> 13.8; 70B: 37.1 -> 31.6, 40.6 -> 36.7).
>
> We would also like to refer to Figure 4(a) in the paper, which shows that, with random execution feedback, models can still propose many solution candidates to the effect that pass@10 scores are closely matched. With execution feedback, solutions proposed in later turns are more targeted, either wrt repairs or in terms of new approaches, which leads to gains in precision (pass@1).
>
>
> > An even more realistic setup in to let LLM self-generate some unit tests, and use it in the multi-turn training process, this might be out of the scope of the current paper, but might be interesting to see.
>
> We agree, a combination with test generation would be very interesting. We'll hopefully see this in future work.

---

### Official Review · Reviewer_o59z · 2025-03-14

**Overall Recommendation:** 3

**Summary:**

This paper finetunes LLMs for multi-turn code generation with PPO. The action is code generation/refinement using LLMs. The model is given public test cases for code evaluation and then refinement. The epsidoe ends either when reaching the maximum turn limit or when the generated codes pass the public tests. The reward of the episode is then if the final codes pass the private the test cases. The model is finetuend and evaluated using the train/val/test splits of CodeContests.

## update after rebuttal

It's a pity that the improvements on livecodebench are not significant.

**Claims And Evidence:**

This paper claims the proposed method, RLEF, is state-of-the-art for improving code generation performance. However, the model is only evaluated on old benchmarks such as CodeContests, HumanEval, and MBPP, suffering the risk of data contamination. Also, there is no baseline performance using Llama-3.1 as the backbone, which raises the concern of unfair comparison. More experimental results are needed to support the claim.

**Essential References Not Discussed:**

N/A

**Experimental Designs Or Analyses:**

The baseline models (as listed in Table 1) all use old pretrained models. Llama 3.1-70B, by itself, already bypasses many of them with RLEF. It is hard to tell if RLEF is the key factor for performance improvement.

The model is only evaluated in old benchmarks as well. It would be much better to evaluate the model in recent benchmarks such as livecodebench.

**Methods And Evaluation Criteria:**

The method is intuitive, using PPO to tune multi-turn code generation models.

**Other Comments Or Suggestions:**

N/A

**Other Strengths And Weaknesses:**

N/A

**Questions For Authors:**

* Are there results in newer benchmarks?
* Are there results of baselines using the same backbone?
* Are there results in more diverse settings, such as more samples per rollout with and without the public test cases?

**Relation To Broader Scientific Literature:**

It targets the important multi-turn code generation problem and shows improved performance in several benchmarks.

**Theoretical Claims:**

No theoretical claims.

---

> ### Author Rebuttal · Authors · 2025-04-01
>
> We thank the reviewer for their valuable feedback and comments and provide the following responses:
>
> > Data Contamination
>
> We build off the Llama 3.1 models which were originally evaluated on benchmarks like HumanEval and MBPP as well. On top, we add training data from CodeContests exclusively; we hence regard the risk of data contamination as low.
>
> > No baseline with Llama 3.1 Backbone
>
> We thank the reviewer for this suggestion. We ran AlphaCodium with Llama 3.1 70B Instruct and obtained solve rates of 34.2 on valid and 27.8 on test, i.e., below the corresponding GPT-4 numbers and notably *below* the 10@100 results for this model, which uses an equal sample budget.
>
> We hence do not think that starting from Llama 3.1 results in an unfair advantage.
>
> > The model is only evaluated in old benchmarks as well. It would be much better to evaluate the model in recent benchmarks such as livecodebench.
>
> We will add a LiveCodeBench evaluation with questions up to 10/2024 in the Appendix, extending Table 2: https://imgur.com/a/CpsZMJB
>
> While we see significant improvements for both model scales, we do not observe gains of the same magnitude as on CodeContests.
>
> > Are there results in more diverse settings, such as more samples per rollout with and without the public test cases?
>
> In Figure 4 we increase the number of samples per rollout (turns), and we also measure performance when iterating on private tests in Appendix B.3 and observe additional (albeit limited) gains.

---

### Official Review · Reviewer_JjJq · 2025-03-14

**Overall Recommendation:** 4

**Summary:**

This paper introduces RLEF - a reinforcement learning method for improving natural language to code generation in an iterative setting. The method treats code generation as a multi-turn conversation, where a language model first produces a program then receives and interprets textual execution feedback to refine solution. This feedback is incorporated as part of the RL training loop. At each turn, the model’s reward depends on whether the final solution passes a private unit tests, while intermediate attempts are guided by test outputs in the prompt based on public tests.

The model is evaluated on the CodeContests competitive programming benchmark showing that RLEF-trained models achieve notably high accuracy using fewer attempts than earlier approaches such as AlphaCode and more recent GPT-based agentic pipelines. The authors provide analyses showing that the model not only generates correct solutions more often but also reliably fixes its own errors in response to execution messages.

**Claims And Evidence:**

Most of the paper’s key claims appear to be backed by concrete evidence. In particular, the central claim that RLEF enables a model to iteratively repair its output and reliably improve code solutions is supported by detailed experiments on CodeContests.

**Essential References Not Discussed:**

AutoCodeRover (Zhang et al., 2024) and other recent automated debugging agents that iteratively fix GH issues using test failures as guidance. While they use tool-based error localization, RLEF’s policy learning approach could complement these methods, making their inclusion relevant to agentic coding workflows.

**Experimental Designs Or Analyses:**

The paper’s experimental setup is well-structured, using CodeContests, HumanEval+, and MBPP+ to test RLEF’s ability to improve NL to code generation through execution feedback. Solve rates, pass@k, and sample efficiency comparisons are appropriate metrics, and multi-turn evaluation reflects real-world debugging scenario. Ablation studies (like e.g. random feedback) confirm that the model genuinely learns from execution feedback rather than random resampling.

The experiments are strong and well-designed, but direct DeepSeek, GPT, etc model benchmarking, real-world coding tasks, and alternative reward structures would further validate RLEF’s generalization and efficiency.

**Methods And Evaluation Criteria:**

The proposed method focuses on iterative code generation and repair, where a model generates an initial solution, receives execution feedback (error messages, test results), and refines its solution. The method also makes sense from an agentic AI perspective, as it pushes models toward self-correcting behavior, which is critical for real-world coding applications.

The evaluation criteria and benchmarks chosen for testing RLEF -- CodeContests, HumanEval+, and MBPP+ -- are reasonable choices, as they represent progressively challenging levels of function synthesis and competitive programming tasks. However, one potential limitation is that these benchmarks focus primarily on single-function correctness and may not fully assess RLEF’s potential for broader software engineering workflows (e.g., multi-file program synthesis, debugging, GitHub issue resolution). While HumanEval+ and MBPP+ demonstrate some generalization beyond competitive programming, additional real-world benchmarks, such as SWE-bench or GitHub Issues, could further validate RLEF’s applicability to industrial coding tasks. Nonetheless, for the specific problem of iterative function-level code improvement, the chosen evaluation framework is sound, and the results convincingly demonstrate the effectiveness of execution feedback-driven RL.

**Other Comments Or Suggestions:**

no

**Other Strengths And Weaknesses:**

Strengths:
* The integration of execution feedback as a dynamic env signal in RL is novel compared to prior one-shot RL for code generation (e.g., CodeRL, AlphaCode).
* Bridges reinforcement learning and agentic AI, making it relevant for self-improving AI coding assistants, debugging agents, and autonomous software maintenance.
* The paper is well structured and easy to read.

Weaknesses:
* Limited evaluation scope: CodeContests is a strong benchmark but focuses on single-function correctness, not full software debugging or open-ended programming tasks. Evaluation on real-world bug-fixing benchmarks (e.g., SWE-bench) would strengthen the paper.
* No direct comparisons to GPT-4, DeepSeek-R1 or other leading models for code.
* RLEF uses binary pass/fail rewards, but no ablation tests whether denser rewards (e.g., partial credit for fixing certain test cases) could accelerate training.

**Questions For Authors:**

1) Explore finer-grained rewards:
RLEF uses a binary pass/fail reward, but no ablation tests whether partial credit for fixing certain test cases or reducing runtime errors could improve the results.
Did you experiment with denser rewards (weighted scoring based on passing test subsets or fixing specific error types)?

2) How well does RLEF generalize beyond function-level synthesis?
The evaluation focuses on competitive programming benchmarks (CodeContests, HumanEval+, MBPP+), which emphasize single-function correctness rather than real-world software debugging or multi-file code synthesis.
Have you tested RLEF on different prompt styles and benchmarks like SWE-bench, or other multi-file programming tasks?
If RLEF does not generalize well beyond competitive programming, its real-world applicability may be more limited than implied. A broader evaluation would solidify its impact for practical software engineering tasks.

3) How does RLEF compare to iterative prompting methods like Reflexion or Self-Refine?
The paper compares against standard fine-tuned baselines but not against iterative self-improvement prompting approaches (e.g., Reflexion, Self-Refine).
Have you tested RLEF against few-shot prompting strategies that also incorporate execution feedback (but without RL-based fine-tuning)?

4) Performance on out-of-distribution tasks
The benchmarks used (CodeContests, HumanEval+, MBPP+) are very well established but contain somewhat templated problem formats.
Have you tested RLEF on entirely novel problem distributions, such as real-world coding tasks and prompt styles not present in training?

**Relation To Broader Scientific Literature:**

The paper builds on reinforcement learning for code generation, drawing from RLHF but replacing human preference signals with automated execution feedback. This aligns with some prior work like CodeRL (Le et al., 2022) and RLTF (Liu et al., 2023), which also optimized LLMs using execution-based rewards. However, RLEF extends this by treating execution feedback as an interactive state rather than a static reward, enabling multi-turn code refinement instead of single-shot optimization.

RLEF also connects to LLM-as-agent frameworks, such as AutoCodeRover (Zhang et al., 2024), which resolves GitHub issues via iterative debugging. Unlike AutoCodeRover, RLEF trains the policy itself, rather than relying on prompting alone. Additionally, it relates to DeepSeek-R1, which also leverages execution-based RL, but RLEF explicitly integrates textual execution feedback into decision-making, making it more akin to self-debugging code models. These connections places RLEF at the intersection of RL-based LLM fine-tuning, execution-grounded AI agents, and automated software repair.

**Theoretical Claims:**

The paper does not introduce new formal theoretical proofs but relies on established RL theory, specifically PPO. The correctness of PPO’s formulation is well-documented, and the paper applies it appropriately without requiring independent verification.

However, the paper makes implicit theoretical claims, such as:
* Multi-turn execution feedback leads to better policy learning than independent sampling, and
* Binary pass/fail rewards are sufficient for meaningful RL-based improvement.

While these claims are empirically validated, they lack formal guarantees. Additionally, exploring whether dense reward shaping (like, penalizing specific error types differently) would improve learning efficiency remains largely an open question.

---

> ### Author Rebuttal · Authors · 2025-04-01
>
> We thank the reviewer for their thoughtful review and valuable feedback.
>
> Based upon their feedback, we investigated the performance of DeepSeek-R1-Distill-Llama-70B in our exact same setting. With a single generation (temp=0.6, prompted with "<think>\n"), we obtained solve rates of 38.5 and 33.9 on the valid and test sets; with execution feedback over three turns, we obtained 41.0 and 37.6, respectively. This places the model in the same ballpark as our RLEF-trained 70B model (37.5 and 40.1).
>
> However, we also note that this required a vastly increased inference budget compared to our models, with up to 10k thinking tokens per turn (this limit would often be reached, and we force-close the thinking section in this case). This renders the comparison quite difficult as our paper places a large emphasis on apples-to-apples comparisons, and allowing for large reasoning budgets would effectively shift the goal posts.
>
> That said, it is evident that DeepSeek-R1 or o1/o3 excel at competitive coding tasks. Therefore, we will update our abstract, intro and experimental work section and do no longer claim state-of-the-art performance in competitive coding tasks.
>
> > AutoCodeRover (Zhang et al., 2024) and other recent automated debugging agents that iteratively fix GH issues using test failures as guidance.
>
> We thank the reviewer for this pointer; we added a reference in our related work section.
>
> > Limited evaluation scope
>
> We fully agree that extending our work to open-ended and long-horizon coding tasks SWE-bench is an exciting future direction of our work. For the current paper, however, we consider this out of scope.
>
> For SWE-bench in particular, current research suggests that substantial domain-specific fine-tuning is required to achieve competitive results. For closed-source frontier models, performance started to increase rapidly once the benchmark was available and the distribution of tasks was known.
>
> We note that we exclusively train on the CodeContest training set, and we would expect that widening the evaluation scope towards new settings will likely come with new training data requirements.
>
> > No direct comparisons to GPT-4, DeepSeek-R1 or other leading models for code.
>
> We discuss DeepSeek-R1 at the top of this reply. We include GPT-4o in Table 2, and GPT-4 is used in AlphaCodium and MapCoder (Table 1) which we outperform.
>
> > RLEF uses binary pass/fail rewards, but no ablation tests whether denser rewards (e.g., partial credit for fixing certain test cases) could accelerate training.
>
> We agree with the author but defer this to further work. Here, we showed that even with a sparse outcome reward we can achieve large gains. We did not experiment with denser rewards.
>
> > RLEF vs. Reflexion or Self-Refine
>
> We refrained from applying prompting strategies in our work and instead focused on domain-specific fine-tuning. However, CodeTree (https://arxiv.org/abs/2411.04329) contains results on the CodeContests test sets with Reflexion (Table 2). With Llama 3.1 8B, they obtain a solve rate of 13.5 with 20 samples. Our RLEF-trained version obtains 16.0 with a budget of 3 samples.
>
> In our paper we also experimented with few-shot prompting (Table 3) but found it to not work well for our use-case.

---

### Official Review · Reviewer_ZGUC · 2025-03-19

**Overall Recommendation:** 4

**Summary:**

This paper proposes an RL training strategy for Code LLMs to enable them to refine generated code using execution feedback besides the capability of following instructions. They present an exhaustive analysis on different aspects of their RLEF-trained models including their inference time behavior, performance gains at different sampling budgets, comparison with other state of the art solutions, and algorithmic alternatives like SFT, single turn RL training, prompting, etc.

**Claims And Evidence:**

I find the evidence presented through the experiments in this paper convincing overall, except for one issue when studying the inference time behavior described below.

- Fig 3 / Line 295: The goal here is to study and establish the sensitivity of RLEF trained model to the feedback from execution. I think using no feedback would have made more sense to be studied as a baseline to compare against, as random feedback included in the prompt to an LLM at any turn is very likely to hurt model performance.

**Essential References Not Discussed:**

I understand techniques like GRPO gained prominence only recently with the release of Deepseek-R1, but I believe adding commentary or comparison to such alternatives to PPO and the requirement of the value function network can benefit the placement of the paper in the present day context of LLM research. E.g. algorithms discussed in Back to Basics: Revisiting REINFORCE Style Optimization for Learning from Human Feedback in LLMs (Ahmadian et al)

**Experimental Designs Or Analyses:**

Yes, I find the experimental design and analyses to be very sound.

**Methods And Evaluation Criteria:**

In my assessment, the proposed methods and evaluation criteria make a lot of sense for the problem of code generation from natural language instructions.

**Other Comments Or Suggestions:**

Suggestion: Discussion on extension to domains beyond code synthesis is only loosely referred to towards the end of Section 4, but the introduction section seems to offer greater promise. For instance, authors could discuss how execution feedback can be generalized to collect beyond code - on verifiable domains like math related tasks, and how all such signals combined (human, execution environment, math correctness) can lead to training of more capable LLM agents. Positioning this contribution in the context of recent advances like Deepseek-R1 will greatly benefit this paper.

Minor/typos:

Line 306: the a --> the

Section 3.2: Para 2: with a single rollout compared to 5 solutions from 100 samples (38.0 and 29). Did you mean 40.1 here instead of 38? Otherwise the results in Table 1 don't match with this description.

Line 244: What does "stock" mean here?

**Other Strengths And Weaknesses:**

Strengths:

- Idea is very well-positioned - motivating RL for training agents that can follow instructions and incorporate feedback from environments

- Adaptation of PPO to the setting of code as described in Section 2.2 is non-trivial
    - Commentary on how they differ from prior work (Le et al (CodeRL)) could help establish their novelty in Section 2 itself, instead of deferring this discussion to the related work.

- Results are convincing
    - CodeContests is a very challenging benchmark, and their baselines include prominent agentic frameworks
    - Authors have augmented their findings with good intuition for readers based on their ablation studies (Section 3.3, Line 281 onwards). I particularly liked the thorough nature of the study performed on inference time behavior.
    - Authors have put in great care in ensuring baselines are reasonable to compare with (for eg - see Line 209)

- Paper is well written and the caveats of the methodology design are sufficiently described

**Questions For Authors:**

- In Section 2.2, you mention the choice of granularity of reward / advantage being action level instead of token level. Can you elaborate more on the alternatives to this choice and what impact one could expect from choosing token level reward or advantage?

- Section 3.2 Para 1: "Each solve rate is estimated on 200 rollouts ..." Here do you not discard the rollouts where the final solution obtained does not pass public tests?

- Table 1: What would be the impact of sampling temperature on these results? Why was greedy decoding not attempted with the multi-turn single rollout setup? (where 1 rollout <= 3 samples/turns?)

- Line 252: To combat the effect mentioned here (RL training reducing diversity of outputs) can one use higher temperature? What could be other ways of addressing this concern? Is this a concern at all?

- Line 286 - you attribute the higher scores in iterative setting to increased diversity in sampled solutions. This seems to contradict with Line 250 where you cite Kirk et al (2024) who find that RL training can reduce diversity also suggested by your results in Table 1. I am confused with the conclusions from these 2 lines.

- Fig 3: Why do you choose to indicate errors fixed in turns 2 and 3? As a reader I think showing the drop of errors over the turns would be easier to follow through this plot.

- Fig 4b: Can you comment on the gap between the instruct and RLEF models? Does it show any upward or downward trend with the sample budget?

- Section 3.4.1 - What is the scale of the SFT dataset synthesized for this study? I'm not certain if suggesting the ineffectiveness of SFT for iterative code refinement is a valid takeaway. I think this finding needs to be augmented with a solid explanation - which could perhaps be the difficulty in designing synthetically / collecting supervised datasets for this task.

- Section 3.4.2 - Is the single turn SFT dataset coming from Code Contests benchmark? Is the takeaway here that this training set is most effective only when used through the RLEF stage (which uses only the test cases and not the ground truth solutions)? Generally SFT datasets are used to fine-tune a checkpoint, and then environment feedback is used to train in the RL phase, but I see no such comparison here. What'd the result be if RLEF was performed on the SFT (single/multi turn) trained checkpoint?

- Section 3.4.2 - "We attribute this to the existent but comparabily weak multi-turn capabilities of the vanilla ..." I don't agree with this conclusion, and I would instead attribute this to the RLEF (ST) training given that the instruct model doesn't benefit much from MT during inference (25.6 to 25.9), whereas after RLEF the gains are a lot more pronounced (28.3 to 31.1).

**Relation To Broader Scientific Literature:**

This work is related to prior work on code generation like AlphaCode, AlphaCodium, Mapcoder and CodeRL.

**Theoretical Claims:**

NA

---

> ### Author Rebuttal · Authors · 2025-04-01
>
> We thank the reviewer for the thorough analysis of our paper, their valuable suggestions and stimulating questions.
>
> ## Updates
>
> Re. GRPO: We will add the following to the end of our related work section:
>
> More recently, DeepSeek-AI et al. (2025) observe emerging reasoning capabilities with a large-scale application GRPO (Shao et al., 2024) to math and code problems and achieve high performance on competitive programming tasks. We thus consider the training of reasoning models with program execution feedback and, likewise, the introduction of execution feedback to math domains, a promising avenue for future research.
>
> > Line 244: What does "stock" mean here?
>
> This refers to the officially released Llama models, in this case Llama 3.1 70B Instruct; we will clarify the wording.
>
> ## Answers to Questions
>
> ### Sensitivity to Execution Feedback
>
> Due to space constraints, we refer the reviewer to our response to Reviewer 3JDi below, where we discuss the choice of random execution feedback and also list solve rates for a "no execution feedback" test.
>
> ### Granularity of Reward: Turn- vs. Token-Level
>
> We provide experiments regarding a token-level value function in Appendix B.5. With token-level rewards, we found that the KL penalty biases the model to unreasonably short generations in intermediate turns, and in B.5 we test the combination in which we still provide averaged KL penalties per turn but a learn token-level value function (and hence obtain per-token advantages). We observe worse results with this approach.
>
> ### Discarding Rollouts
>
> We generally do not discard rollouts for evaluation and do not consider public tests in scoring unless noted otherwise. A rollout with 3 turns without a solution passing the private tests will count as a failed sample, similar to rollouts with 1 or 2 turns where public tests are passing. A rollout with a successful result will count as a successful sample. In other words, one rollout corresponds to one sample.
>
> To compare under equal sampling budgets, we then consider the "1@1" and "1@33" performances in this setting as "1@3" and "1@100" solve rates. NB, models will not utilize the full budget due to some rollouts ending early with correct responses.
>
> ### Results with Greedy Decoding
>
> We observe slightly better results with a low temperature compared to greedy decoding.
>
> ### Diversity in Outputs
>
> The question of whether a loss of diversity is a concern depends on the intended applications. Remedies can be found on the RL algorithm level as shown in https://arxiv.org/abs/2503.19595, for example. We found a temperature of 1.0 to deliver best performance in the large-sample regime; this may be linked to the fact that we use a temperature of 1.0 during rollouts as well.
>
> Re Line 286, output diversity depends on the evaluation setting. From the analysis in Table 3 it is clear that base models do not sample diverse solutions *within* a rollout. However, they can produce a large number of different solutions from the initial prompt, in particular with higher temperatures. What makes RLEF-trained models effective is that they can utilize execution feedback to either repair an existing solution or to propose a new approach.
>
>
> ### Figure 3
>
> We selected to show "errors fixed" here to highlight differences between the different models and settings. The X-axis scale is different for "Errors" and "Errors Fixed". While we regard the reported differences as significant, they would be hard to discern visually under the X-axis scale of the leftmost plots.
>
>
> ### Gap Between Instruct and RLEF models wrt. Sample Budget
>
> We did not study the diversity aspect in the limit, i.e., under very large sample budgets. We would expect the gaps to narrow in the limit due to decreased output diversity. An interesting question would however be if this can be offset by sampling longer and longer trajectories.
>
>
> ### SFT Dataset
>
> Yes, the SFT dataset was obtained solely from the CodeContest training dataset. It consists of 313,639 trajectories. The result that RL can work better than SFT is in line with references discussed in the paper, i.e., Xu et al and Kirk et al, and also the recent results around DeepSeek-R1.
>
>
> ### Existing MT capabilities of 70B
>
> > Section 3.4.2 - "We attribute this to the existent but comparabily weak multi-turn capabilities of the vanilla ..." I don't agree with this conclusion, and I would instead attribute this to the RLEF (ST) training given that the instruct model doesn't benefit much from MT during inference (25.6 to 25.9), whereas after RLEF the gains are a lot more pronounced (28.3 to 31.1).
>
> We were referring to the fact that the 70B model already exhibits multi-turn capabilities (although the valid set difference is not statistically significant) when compared to the 8B model, for which performance drops in the multi-turn setting.

---

### Decision · Program_Chairs · 2025-05-01

**Decision:**

Accept (spotlight poster)

**Comment:**

In this paper, the authors propose an end-to-end RL method to train models to learn from execution feedback for code generation tasks. The authors conducted comprehensive experiments on different model sizes (from 8B to 70B parameters) and demonstrated significant performance gains from the proposed method against strong baselines. The authors include good ablation study (e.g. inference-time behavior) with useful insights for the community. There are some minor issues raised by the reviewers, including the application to more practical tasks e.g. SWEBench, and comparison to more advanced LLMs such as GPT and DeepSeek.

Overall, I found the proposed method a well designed and elegant solution in the intersection of RL for LLM, AI agentic framework, and self-refining/ self-debugging code generation. I suggest the authors review and incorporate all the feedback into their final revision.